# Tufas indicate prolonged periods of water availability linked to human occupation in the southern Kalahari

Jessica von der Meden[1,2]*, Robyn Pickering[1,2], Benjamin J. Schoville[2,3], Helen Green[4], Rieneke Weij[1,2,4], John Hellstrom[4], Alan Greig[4], Jon Woodhead[4], Wendy Khumalo[1,2], Jayne Wilkins[2,5]

1 Department of Geological Sciences, University of Cape Town, Rondebosch, South Africa, 2 Human Evolution Research Institute, University of Cape Town, Rondebosch, South Africa, 3 School of Social Science, University of Queensland, St Lucia, Queensland, Australia, 4 School of Geography, Earth and Atmospheric Sciences, The University of Melbourne, Parkville, Victoria, Australia, 5 Australian Research Centre for Human Evolution, Environmental Futures Research Institute, Griffith University, Nathan, Australia

* vdrjes001@myuct.ac.za

**Data Availability Statement:** All relevant data are within the article and its Supporting Information files.

## Abstract

Detailed, well-dated palaeoclimate and archaeological records are critical for understanding the impact of environmental change on human evolution. Ga-Mohana Hill, in the southern Kalahari, South Africa, preserves a Pleistocene archaeological sequence. Relict tufas at the site are evidence of past flowing streams, waterfalls, and shallow pools. Here, we use laser ablation screening to target material suitable for uranium-thorium dating. We obtained 33 ages covering the last 110 thousand years (ka) and identify five tufa formation episodes at 114–100 ka, 73–48 ka, 44–32 ka, 15–6 ka, and ~3 ka. Three tufa episodes are coincident with the archaeological units at Ga-Mohana Hill dating to ~105 ka, ~31 ka, and ~15 ka. Based on our data and the coincidence of dated layers from other local records, we argue that in the southern Kalahari, from ~240 ka to ~71 ka wet phases and human occupation are coupled, but by ~20 ka during the Last Glacial Maximum (LGM), they are decoupled.

## Introduction

A key question in human origins research is how climate change impacted early *Homo sapiens* population distributions across Africa. It has been hypothesized that humans did not always have the capacity to survive in arid environments [1, 2], that early human distributions were modulated by distance to [3] and availability of water [4], that people were largely restricted to wetter refugia during glacial periods [5–7], and that the occupation of arid regions was coincident with interglacial periods [8, 9]. We see this view manifested slightly differently in different regions of the African continent. For example, in eastern Africa, favourable climatic conditions and increasing water availability during MIS 5 is associated with greater mobility of *Homo sapiens* and their occupation of a wide variety of sites, not restricted to oases such as lake margins [7]. In North Africa, the idea of a 'Green Sahara' during ~130–75 ka is contested by Scerri et al. [10] who use a palaeoenvironmental model to highlight that the region was not a uniform

**Funding:** Funding for this project was provided by the Department of Science and Technology National Research Foundation (DST-NRF, South Africa) Centre of Excellence in Palaeosciences through student bursaries (JvdM and WK) and Operations grants - CoE2017-065, COE2018-05OP, COE2019-OP17 and COE2018-10OP (JWi and RP); the National Research Foundation (NRF, South Africa) African Origins Platform Grant - AOP150924142990 (RP), NRF (South Africa) Competitive Programme for Rated Researchers -120806 (RP), National Geographic Society - Waitt Grant (BJS), University of Cape Town VC2030 funding (RP), NRF (South Africa) Research Development Grant for Y-rated Researchers (JWi), and Australian Research Council Discovery Early Career Research Award - DE 190100160 (JWi). The funders had no role in study design, data collection and analysis, decision to publish, or preparation of the manuscript.

**Competing interests:** The authors have declared that no competing interests exist.

oasis, noting that pockets of aridity persisted as barriers to mobility, even during periods of increased humidity, and that human dispersal at this time occurred along corridors connected by palaeohydrological networks. While evidence for human mobility and dispersal is generally linked to water availability [10, 11], there are some exceptions, for example, occupation at the North African site of Uan Tabu between ~60–90 ka is associated with evidence of an arid climate [12]. With recent advances in the application of dating techniques e.g. U-Th and OSL methods, to archaeological and palaeoclimate proxy deposits, improved chronological reliability and resolution allows for questions relating to human-environment interactions to be refined [13, 14].

The Kalahari Basin, in the interior of southern Africa, is a semi-arid region that has experienced significant climatic fluctuations with abundant records of both palaeoenvironment and archaeology. As such, this region provides a unique opportunity to further explore early human-environment interactions [15]. For example, in the southern Kalahari, multiple lines of evidence point to very different, much wetter periods during much of the Pleistocene [16]. Paleoenvironmental records from macrobotanical and faunal remains, pollen, and stable isotope compositions of mammalian tooth enamel, ostrich eggshells and speleothem deposits, demonstrate climatic shifts through the Pleistocene and Holocene at Wonderwerk Cave [17–24]. At Kathu Pan, several wet periods between ~160–22 ka have been identified based on sedimentary analyses [25, 26], and evidence for wetter conditions at Ga-Mohana Hill ~110–105 ka has also been reported [27]. Previous studies reveal significant complexities even at the intra-regional scale, however, due in part to the different types of proxies with variable resolutions and the variety of forcing factors at play [16]. They also reveal a complex relationship between palaeoenvironmental conditions and evidence for human occupation [15, 28]. To more fully assess the response of *Homo sapiens* to changes in climate and environments, more well-dated records of past environments from different proxies, preferably closely associated with archaeological records of human behaviour, are required. Here, we report one such record from the abundant carbonate deposits at Ga-Mohana Hill, identified as tufa i.e., ambient temperature, freshwater calcium carbonate precipitates, that span the last 110 thousand years.

Ga-Mohana Hill is a double-humped hillside situated on the eastern flank of the north-south trending Kuruman Hills which outcrop on the Ghaap Plateau, an elevated region in the Northern Cape province of South Africa (Fig 1). Today the area is characterised as semi-arid, with seasonal mean annual precipitation of ~300–400 mm during the austral summer months [24]. The bedrock lithology of the Ghaap Plateau, which is comprised of the Palaeoproterozoic dolomites of the Campbellrand-Malmani Subgroup [29], has undergone extensive karstification. Coupled with cross-cutting dolerite dykes, this has resulted in large groundwater compartments that are important aquifers for the region [30, 31]. Groundwater resurgence at active springs in the area, such as the Eye of Kuruman, in Kuruman (Fig 1) are a testament to this vast underground drainage network [31]. The presence and movement of these groundwaters through the dolomite host rock is a vital precursor to the formation of the tufas at Ga-Mohana Hill.

Recent archaeological excavations at Ga-Mohana Hill North Rockshelter have yielded a Middle Stone Age assemblage of artefacts that provides early evidence for innovation and behavioural complexity in this region at 105±3.3 ka [27, 36]. Stratified above are younger deposits dated by OSL to 31±1.8 ka and 15±0.8 ka [36]. The hillside has abundant tufa deposits which are direct evidence for the presence of water on the landscape and are amenable to radiometric dating methods, making them valuable archives of changes in environmental conditions [37–42].

In this study, we present macro- and micromorphological analyses of the Ga-Mohana tufas to assess their depositional context. Tufas are challenging materials for dating due to detrital

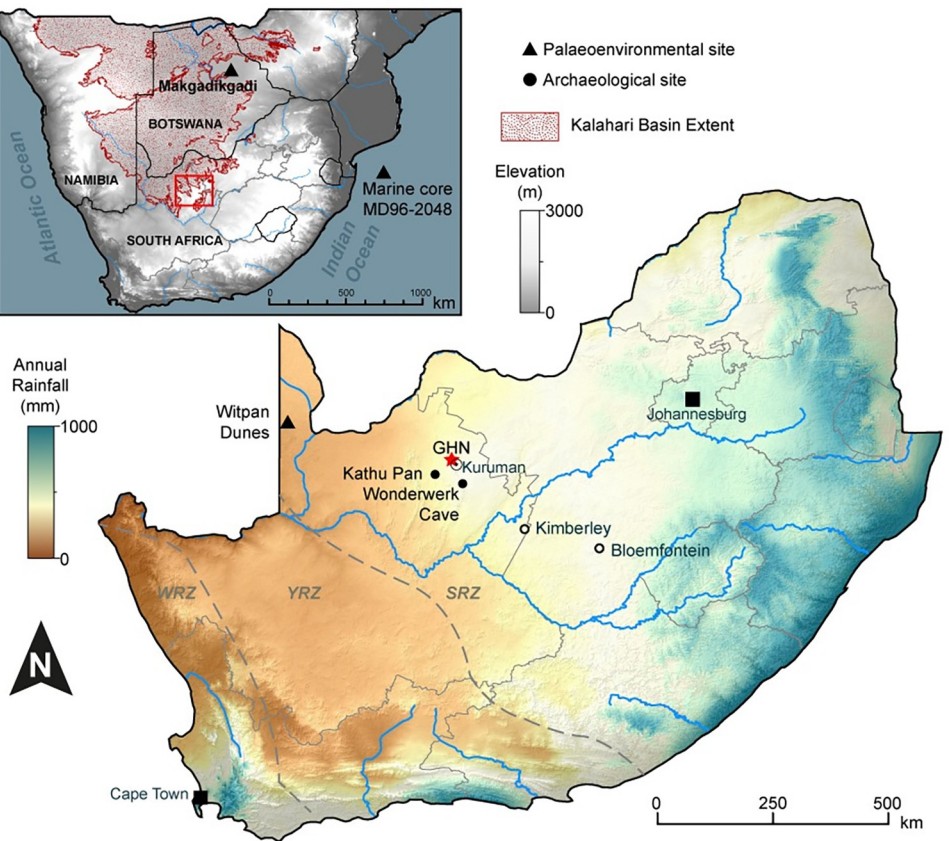

**Fig 1. Map of South Africa with the location of Ga-Mohana Hill (GHN) and key palaeoenvironmental and middle stone age sites discussed in the text.** Dashed lines demarcate summer and winter rainfall zone boundaries (SRZ, WRZ), middle area experiences year-round rainfall (YRZ). Inset map shows the approximate extent of the Kalahari Basin in southern Africa and the location of the region of interest in relation to it. Figure produced in ArcGIS 10 from multiple open source datasets: Kalahari Basin extent from SASSCAL Open Access Data Centre [32]; Digital Elevation Model obtained from USGS Earth Explorer [33]; annual rainfall data from WorldClim 2 [34] and river centrelines accessed from Natural Earth vector data [35].

contamination and generally low uranium concentrations [43, 44] and so samples were screened using laser ablation inductively coupled plasma mass spectrometry (LA-ICP-MS) to target optimal zones for study. This method has been used previously for dating speleothems [45], but to the best of our knowledge, this is the first tufa application. We present 33 U-Th age estimates and identify five discrete periods of tufa formation, interpreted as evidence for increased moisture and fresh water availability on the landscape during the late Pleistocene, thus providing a new record of localized climate change linked to a dated record of human occupation.

## Materials and methods

### Fieldwork and tufa sample collection

Ga-Mohana Hill has spiritual significance for the local communities, with visits to the shelter deliberate and rare [46]. Out of respect for this and as part of our on-going engagement with these communities, we adopted a low-impact sampling approach, with targeted samples carefully chosen after extensive survey of the 6 km area around the shelter. During this pedestrian

survey, the field occurrences, positions and types of tufa were identified and mapped using a roaming Geographic Positioning System. A total of twenty-nine tufa hand samples were collected from the ~ 1 $km^2$ Ga-Mohana hillside sampling all five tufa morphologies. Eighteen hand samples were collected using a geological hammer, mallet and chisel, marking the way-up on each sample with an arrow using permanent marker. Subsequent sampling deliberately targeted the stratigraphically older layers, closest to the host rock dolomite, to try and constrain the onset of preserved tufa formation. We used a modified Makita cordless hand drill fitted with Pomeroy Model SW-3 Miniature Water Swivel and a custom made Pomeroy 1.5" ID diamond-tipped core barrel. A total of eleven small cores were collected, 8 cm in length on average, from *in-situ* mound tufas, and both *in-situ* and *ex-situ* cascade tufas (S1 Table, S1–S5 Figs). The cores were set in epoxy resin and then halved lengthways with a diamond rock saw and polished. Thin sections were made from a sub-set of fourteen samples, representative of all the morphology types, for characterisation using a Zeiss AXIO polarising light microscope (S1 Table, S1–S5 Figs).

Permits for archaeological investigations at Ga-Mohana Hill were obtained from the South African Heritage Resource Agency (Permit ID 2194). The land is owned by the Baga Motlhware Traditional Council and consent was granted by them to conduct the study. All necessary permits were obtained for the described study, which complied with all relevant regulations. Additional information regarding the ethical, cultural, and scientific considerations specific to inclusivity in global research is included in the Supporting Information (S1 Checklist).

## Laser ablation-inductively coupled plasma-mass spectrometer (LA-ICP-MS) pre-screening of U and Th concentrations and distributions

The aphanitic micrite layers free from detritus and inclusions, identified in thin section, were primary targets for U-Th dating. However, these layers tend to be fine, undulating and laterally variable, and so while visual evaluation of the tufas is an important first step in identifying suitable material to target for U-Th dating, it is not sufficient considering the complexity of the tufas on a microscale. We employed an additional screening step, using laser ablation inductively coupled plasma mass spectrometry (LA-ICP-MS), to measure and image the U and Th concentrations and distributions along transects within the tufa samples. Thus, layers with sufficiently high levels of $^{238}U$ and low levels of $^{232}Th$, i.e. detrital thorium, can be selected, as these are the best for producing reliable age data [47].

Tufa U and Th concentrations and distributions were collected for 16 samples using laser-ablation with an Applied Spectra RESOlution SE 193nm ArFexcimer laser-ablation system coupled to an Agilent 7700x Quadrupole ICP-MS at the University of Melbourne, following the protocols outlined in Woodhead et al. [48]. High-resolution images (3200 dpi) of the samples were captured using a flat-bed scanner, used to reference the co-ordinate system of the laser cell using GeoStar software (Norris Software). Between 6 and 12 parallel lines per sample, set 62μm apart, were chosen perpendicular to the growth layers. Pre-ablation was performed twice using a 60μm spot size and stage translation speed of 150μm/s.

Trace element data for the following elements: Mg, Al, Mn, Fe, Zn, Sr, Ba, Pb, Th and U, were collected with a 60μm spot at a stage translation speed of 75μm/s, pulse rate of 10Hz, and laser fluence of ~2–3 $Jcm^{-2}$. NIST SRM 612 was used for calibration, with $^{43}Ca$ as an internal standard, and an estimated precision of ca <5%. NIST SRM 610 and JCp-1, a powdered coral standard, were also analysed. The raw mass spectrometry data was reduced using the Iolite software package [49, 50]. Element distribution maps for $^{238}U$ and $^{232}Th$ were generated in order to visualise the spatial arrangement of these trace elements through the samples [51] (S6–S8 Figs).

## U-Th dating of tufa

Guided by the laser ablation results, layers with sufficiently high uranium ($^{238}U > 0.1$ppm) and low thorium concentrations ($^{232}Th < 0.01$ppm) were selected for U-Th analysis. A subset of 43 samples (S2 and S3 Tables, S6–S8 Figs), each with a mass of 60 ± 10 mg, were drilled from 16 tufa samples using a Dremel hand-held hobby drill and 1 mm carbide micro-drill bit. Powdered samples were dissolved in 1.5M $HNO_3$, spiked with a mixed $^{236}U$-$^{233}U$-$^{229}Th$ tracer equilibrated on a hotplate overnight. U and Th were separated from the calcite matrix using Eichrom TRU-spec selective ion exchange resin following established protocols [52]. The U-Th solution was dissolved in a mixture of dilute nitric and hydrofluoric acid and introduced to the Nu Instruments Plasma Multi Collector-Inductively Coupled Plasma-Mass Spectrometer via an autosampler [52, 53]. Isotope-ratio measurements for $^{230}Th/^{238}U$ and $^{234}U/^{238}U$ were calculated using an internally standardised parallel ion-counter procedure and calibrated against the secular equilibrium standard, HU-1. Reproducibility was monitored using a second in-house standard (YB-1). An a *priori* estimate of 1.5 ± 1.5 for the initial $^{230}Th/^{232}Th$ was applied to all the samples in order to correct for the inherent detrital component [47]. With this initial value and its uncertainty, corrected ages for all samples were calculated using Monte Carlo iterations to solve equation 1 of Hellstrom [47] and the half-life values of $^{234}U$ and $^{230}Th$ as reported in Cheng et al. [54]. The final age uncertainty is reported in 2σ for which the uncertainties of the measured activity ratios as well as the assumed initial $^{230}Th/^{232}Th$ are fully propagated.

## Results

### Tufa macro and micromorphology

The Ga-Mohana tufa system comprises five morphological components: cascades, rim pools, barrages, domes and terrace breccias (Fig 2) drawing from established classification schemes [55–58]. Tufa cascades are observed across the ~1 $km^2$ hillside. Large (3-5m) cascades cover the tall cliffs on either side of the rock shelters. Smaller cascades bulge outward from the fronts of the dolomite steps above and below the shelters (Fig 2, S1 Fig). These cascade tufas are point-sourced, appearing to have formed from water flowing out of the dolomite bedding planes. Below the step-front cascades, sinuous tufa rims edge the flat, transverse sections of the dolomite steps. These are evidence of terraced, shallow pools, likely formed from excess water ponding below the cascades. The areas behind the rims are filled with lightly compacted sediment and debris. Curved, down-hill sloping barrage tufas, characterised by knobbly, coralloid surfaces, sit below the rim pool edges formed when water overflowed from the pools above. Meandering channels scoured in the dolomite, observed above the rock shelter, are evidence of palaeostreams and indicate periods of substantial and prolonged water flow.

The rockshelters and the tall cliffs adjacent to them mark a break in the hillside. At this point, cone-shaped tufa 'noses' jut out over the lip of the rock shelters and are spread along the overhang (Fig 2). These are interpreted as remnants of moss curtains and align with large hemispheric dome tufas below. The domes trace the dripline of the overhanging shelters and occur along the base of the cliffs adjacent to the rock shelters; they appear to be sourced from dripping and splashing waters channelled via the noses above and are composed of stacked phytoherm macro-layers reminiscent of bryophyte boundstones [55, 56]. The internal rock-shelter walls are covered with clusters of small stalactitic features and calcitic crusts. Below the shelter, surface-cementations of sub-angular detrital clasts of variable sizes (0.5–20 cm) of banded ironstone and dolomite fragments occur as benches, pavements, or patches of carbonate-cemented hill-slope material on the sub-horizontal terraces between dolomite steps,

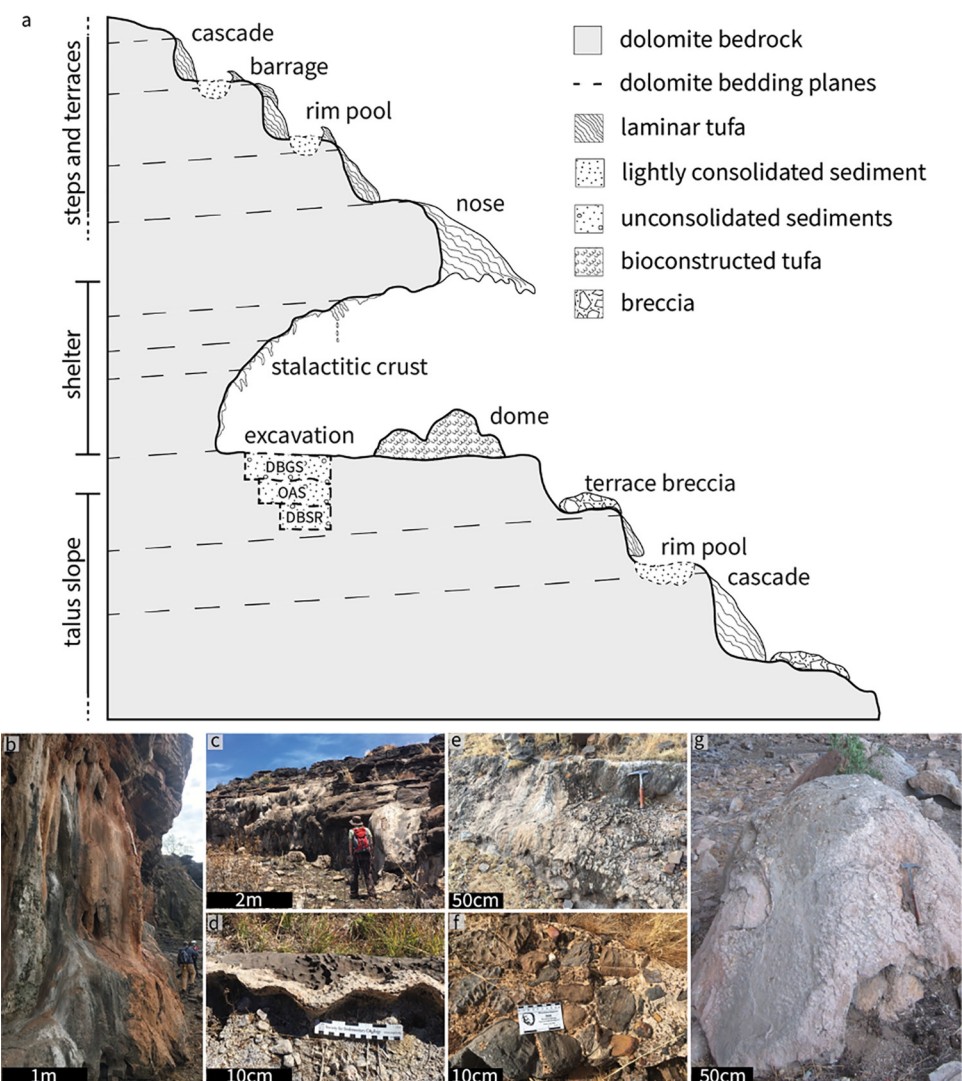

**Fig 2. Tufa depositional environment context and representative photographs of each of the tufa morphologies identified on the Ga-Mohana hillside.** (**A**) Schematic profile sketch of Ga-Mohana Hill North Rockshelter (not drawn to scale) illustrating the series of tufa deposits and the archaeological excavation. The excavation layers dated via OSL: DBSR = ~105 ka; OAS = ~31 ka and DBGS = ~15 ka; (**B**) cliff cascade; (**C**) step-front cascade; (**D**) sinuous rim-pool edge; (**E**) barrage tufa; (**F**) terrace breccia; (**G**) tufa dome.

similar to the surface-cemented rudites described in Pentecost and Viles [58]. In rare instances, stone artefacts are also included.

Microscale observations of the tufas reveal that, regardless of depositional setting, the tufas are composed of a few simple petrographic components: micrite, microspar, and sparite (S1–S5 Figs). Detrital clasts (quartz) and iron and manganese oxides are present to varying degrees and tend to be concentrated along thin layers in the tufas. This suggests periods of non-deposition of tufa. The variable organisation of these petrographic components within each sample results in distinct fabrics, classed as laminar, peloidal, aphanitic, and chaotic, following the scheme devised by Manzo et al. [59]. A significant biological component is evidenced by stromatolitic structures (S1 Fig), clotted micrite (S3 Fig), and primary cavities (S5 Fig).

The various tufa morphologies at Ga-Mohana Hill each represents an individual sub-environment, and together they form a continuum of linked deposits that aligns with the perched springline model of Pedley [57, 60]. This depositional environment was characterised by water emerging from bedding planes in the dolomite, flowing down the hillside via multiple divergent pathways, creating cascades on the step-fronts of the dolomite steps, generating waterfalls and moss curtains over the rock shelters, and feeding shallow pools on the flat terraces. The terrace breccia deposits hint at periods of high energy flow (e.g., flash flooding) to transport and cement substantial talus scree downslope.

## U-Th chronology

The $^{238}$U concentration in the tufas is consistently low (range = 0.1 to 0.6 ppm; mean = 0.2 ppm). The $^{232}$Th concentrations were generally lower than the $^{238}$U concentrations, with most samples reflecting a wide range in $^{232}$Th concentration, between 1–100 ppb. In many instances, elevated $^{232}$Th corresponds with visually discernible detrital material (S8 Fig). Out of 43 sub-samples drilled from 18 tufa samples, we obtained 33 U-Th ages from 12 tufas (Table 1, S2 Table). Cascade, rim pool and terrace tufas exhibited high success rates; 86% of cascade samples (18 of 21), 100% of rim pools (7 of 7) and 100% of terrace breccias sampled (7 of 7) yielded resolvable U-Th age estimates, while only one of four dome samples returned a reliable age. Reliable ages tended to be unresolvable on samples with very low $^{230}$Th /$^{232}$Th ratios (e.g. $^{230}$Th /$^{232}$Th <7) indicating a significant detrital component (S3 Table). It was not possible to resolve reliable or precise ages for any of the barrage samples, three dome and two cascade samples, however some of the corrected ages for these samples may provide a useful upper limit age estimate, i.e. the corrected age plus the associated 2σ uncertainty (S3 Table).

The tufa ages span the last interglacial cycle, from 110.6 ± 3.0 ka through to 3.0 ± 0.9 ka (Table 1, Fig 3). Clusters of ages, defined by distinct groups of overlapping ages and their associated 2σ uncertainties, suggests episodic growth over this time. At least five intervals of tufa formation at Ga-Mohana Hill identified at approximately 114–100 ka, 73–48 ka, 44–32 ka, 15–6 ka, and ~3 ka (Fig 3). The 2σ uncertainties associated with the ages are small; most samples are associated with errors of <3 ka (on average approximately 1 ka) except for two samples, GHS-5 and GHS-6.3, which have an uncertainty of 4.9 ka (49%) and 4.2 ka (7.9%) respectively. These larger errors are due to a high detrital thorium component (Table 1).

The ages for the timing of human occupation at Ga-Mohana Hill coincides with three of the tufa forming intervals during MIS 5d, late MIS 3, and late MIS 2, indicating contemporaneous human activity and tufa formation at Ga-Mohana during those periods (Fig 3). The age certainty for the interval of tufa formation that overlaps with the MIS 2 occupation at Ga-Mohana Hill is less secure than the other intervals as it has a large error associated with it, but the human occupation falls within the 2σ uncertainty of the tufa age.

## Comparison to global records

Tufa deposits occur in a variety of settings around the world, and their formation is controlled by a range of local, regional and global factors operating on different scales [38, 60, 64–66]. Here we compare the timing of tufa formation at Ga-Mohana Hill to records of global ice volume [61], austral summer insolation for $27^0$ south [63], and changes in sea surface temperature in the southwest Indian Ocean [62] to assess the extent to which the tufa deposits reflect global-scale climate changes (Fig 3). There is no clear glacial/interglacial partitioning of tufa formation episodes, as evidenced by comparing our data to the LR04 d$^{18}$O benthic stack [61] (Fig 3). This adds to growing evidence that the wet/dry, interglacial/glacial dichotomy through which much of southern African palaeoclimates has traditionally been viewed is overly

**Table 1. U-Th age data for tufa samples from Ga-Mohana Hill.** The samples are labelled according to the sequence they were collected in but presented in stratigraphic order. Errors on all isotope activity ratios are reported with 2σ uncertainty. All ages have been corrected to account for the effect of detrital Th assuming an estimate for initial $^{230}Th/^{232}Th$ of 1.5 ± 1.5, and calculated using the $^{230}Th$-$^{238}U$ decay constants of Cheng et al. [54] and equation 1 from Hellstrom [47].

| Sample ID | Tufa type | $^{238}U$ (ng/g) | $^{230}Th/^{238}U$ | 2σ | $^{234}U/^{238}U$ | 2σ | $^{230}Th/^{232}Th$ | U-Th age (ka) | 2σ | % error |
|---|---|---|---|---|---|---|---|---|---|---|
| 18–10.2 | dome | 174 | 0.088 | 0.001 | 2.443 | 0.006 | 6.5 | 3.0 | 0.9 | 30 |
| GHN-2 | cascade | 75 | 0.184 | 0.002 | 2.728 | 0.009 | 36.2 | 7.266 | 0.315 | 4.3 |
| GHS-5 | cascade | 263 | 0.251 | 0.002 | 1.866 | 0.007 | 4.6 | 10.738 | 4.932 | 45.9 |
| 17–8.1 | terrace | 363 | 0.707 | 0.002 | 1.895 | 0.003 | 88.4 | 48.306 | 0.684 | 1.4 |
| 17–8.2 | terrace | 273 | 0.528 | 0.002 | 1.894 | 0.003 | 238.3 | 34.503 | 0.231 | 0.7 |
| 17–8.3 | terrace | 321 | 0.621 | 0.002 | 1.896 | 0.005 | 3367.7 | 41.760 | 0.220 | 0.5 |
| 17–8.4 | terrace | 298 | 0.641 | 0.003 | 1.903 | 0.005 | 4516.0 | 43.230 | 0.270 | 0.6 |
| 17–8.5 | terrace | 307 | 0.618 | 0.003 | 1.895 | 0.005 | 4858.7 | 41.620 | 0.260 | 0.6 |
| 17–8.6 | terrace | 459 | 0.503 | 0.004 | 1.899 | 0.006 | 216.6 | 32.450 | 0.370 | 1.1 |
| GHN-1 | rim pool | 236 | 0.576 | 0.005 | 1.863 | 0.006 | 716.3 | 39.126 | 0.398 | 1.0 |
| GHN-1.2 | rim pool | 234 | 0.543 | 0.005 | 1.859 | 0.006 | 6818.5 | 36.550 | 0.390 | 1.1 |
| GHN-1.3 | rim pool | 240 | 0.551 | 0.005 | 1.868 | 0.007 | 10846.4 | 37.020 | 0.420 | 1.1 |
| GHS-6 | rim pool | 212 | 0.864 | 0.007 | 1.914 | 0.007 | 41.3 | 60.379 | 1.809 | 3.0 |
| GHS-6.1 | rim pool | 180 | 0.852 | 0.005 | 1.913 | 0.007 | 50.3 | 59.677 | 1.461 | 2.4 |
| GHS-6.2 | rim pool | 190 | 0.873 | 0.003 | 1.928 | 0.0054 | 251.3 | 61.986 | 0.466 | 0.8 |
| GHS-6.3 | rim pool | 158 | 0.798 | 0.003 | 1.882 | 0.004 | 15.9 | 53.100 | 4.200 | 7.9 |
| 18–7 | terrace | 847 | 0.749 | 0.003 | 1.833 | 0.005 | 50.5 | 53.520 | 1.310 | 2.4 |
| 18–13.1 | cascade | 249 | 1.174 | 0.003 | 2.654 | 0.007 | 68.3 | 58.610 | 0.990 | 1.7 |
| 18–13.2 | cascade | 226 | 1.313 | 0.003 | 2.742 | 0.007 | 93.6 | 65.040 | 0.810 | 1.2 |
| 18–13.3 | cascade | 132 | 1.287 | 0.004 | 2.646 | 0.007 | 620.5 | 67.150 | 0.380 | 0.6 |
| 18–13.4 | cascade | 195 | 1.483 | 0.004 | 2.933 | 0.008 | 124.6 | 69.830 | 0.680 | 1.0 |
| 18–14.1 | cascade | 83 | 1.308 | 0.009 | 2.644 | 0.009 | 243.4 | 68.430 | 0.730 | 1.1 |
| 18–14.2 | cascade | 139 | 1.219 | 0.006 | 2.551 | 0.008 | 46.1 | 64.280 | 1.600 | 2.5 |
| 18–14.3 | cascade | 98 | 1.351 | 0.008 | 2.705 | 0.009 | 439.1 | 69.350 | 0.670 | 1.0 |
| 18–14.4 | cascade | 180 | 1.481 | 0.006 | 2.876 | 0.008 | 47.0 | 70.600 | 1.670 | 2.4 |
| 18–15.1 | cascade | 137 | 1.319 | 0.007 | 2.668 | 0.008 | 781.0 | 68.520 | 0.570 | 0.8 |
| 18–15.2 | cascade | 95 | 1.317 | 0.011 | 2.587 | 0.010 | 746.9 | 71.340 | 0.890 | 1.2 |
| 18–15.3 | cascade | 313 | 1.522 | 0.005 | 2.940 | 0.008 | 229.7 | 72.280 | 0.530 | 0.7 |
| 18–17.1 | cascade | 154 | 2.176 | 0.006 | 3.194 | 0.006 | 29.0 | 102.900 | 3.200 | 3.1 |
| 18–17.2 | cascade | 148 | 2.085 | 0.007 | 3.102 | 0.006 | 44.2 | 102.100 | 2.100 | 2.1 |
| 18–17.3 | cascade | 142 | 2.217 | 0.006 | 3.289 | 0.006 | 95.7 | 103.310 | 1.080 | 1.0 |
| 18–16.1 | cascade | 164 | 2.586 | 0.008 | 3.614 | 0.007 | 32.9 | 110.600 | 3.000 | 2.7 |
| 18–16.2 | cascade | 177 | 2.404 | 0.007 | 3.476 | 0.007 | 43.9 | 105.900 | 2.200 | 2.1 |

simplistic [67–71]. While tufas have typically been associated with warmer and more humid interglacial climate conditions [72–75], several studies report tufa occurrences during both glacial and interglacial periods [41, 42, 66, 76, 77]. This highlights that, across the globe, regional climates respond variably to these boundary conditions, and cautions against simplistic interpretations of tufa deposits. Our record suggests tufa formation was semi-continuous across MIS 4 and MIS 3; we thus echo the conclusion of previous studies that tufa formation is not restricted to interglacial periods, nor is it a simple product of changing global climate states.

The principal conditions required for tufa formation are sufficient effective precipitation to recharge the aquifers and $CaCO_3$ supersaturation of those waters [38, 40, 75]. Productive soil and vegetation cover is necessary to enhance the $pCO_2$ of the percolating waters, and moderate temperatures which balance productivity, moisture and evaporation, are important secondary

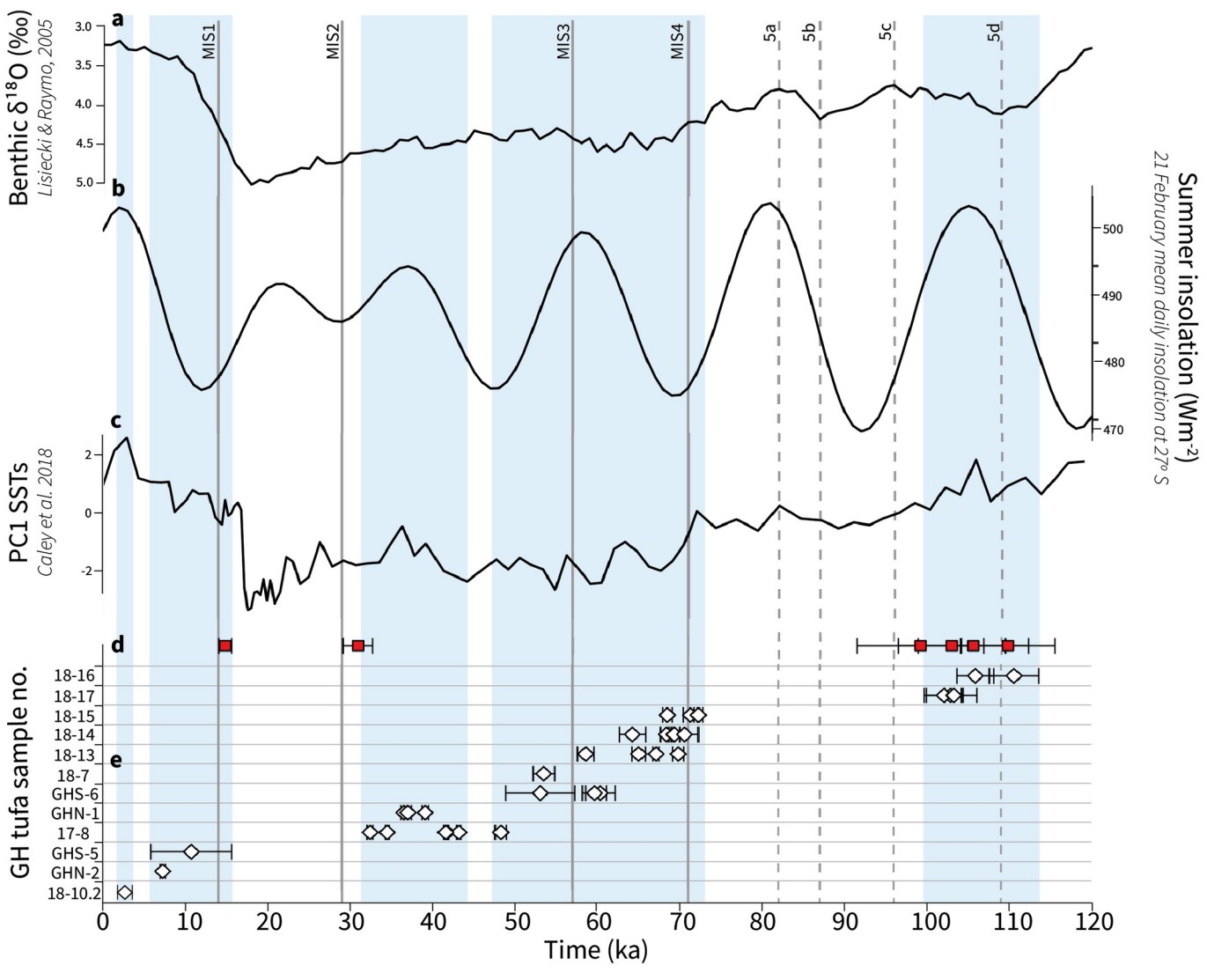

**Fig 3. Composite plot of Ga-Mohana Hill tufa formation compared to selected global proxies over the last 120 ka.** (**A**) LR04 curve [61]; (**B**) variance of reconstructed sea surface temperatures (SST) from Indian Ocean core MD96-2048 [62]; (**C**) mean daily summer insolation curve for 27°S [63]; (**D**) OSL age data from the Ga-Mohana Hill North excavation sediments [27, 36]; (**E**) tufa U-Th age data with 2σ error bars presented in Table 1. The blue bars highlight clusters of overlapping tufa ages and are defined by the minimum and maximum ages in each range, calculated using the 2σ uncertainty associated with the ages. Based on the presence of the tufa deposits, these periods are inferred to represent episodes of greater water availability on the landscape.

requirements [38, 42]. Tufa formation is thus sensitive to multiple environmental parameters, but ultimately provides direct evidence of fresh water and associated productivity on the landscape. Our record indicates that these conditions were met during five discrete time intervals (114–100 ka, 73–48 ka, 44–32 ka, 15–6 ka, and 3 ka) in the southern Kalahari over the last ~110 ka.

The limiting factor for tufa formation in semi-arid, low latitude regions is water availability [40, 78]. The moisture source for rainfall in the SRZ in South Africa originates primarily from

the south western Indian Ocean [79] but the spatial and temporal variability of rainfall in this southern Kalahari region is poorly constrained. Rainfall may be modulated by summer insolation, with increased precipitation corresponding to insolation maxima [80], however, we see no simple correlation. Based on the mean summer insolation curve for 27˚S [63] (Fig 3), tufa formation during the 114–100 ka and 44–32 ka intervals coincide with increasing summer insolation, while tufa formation during 73–48 ka is variable, and at a minimum during the 15–6 ka episode. The most recent tufa formation at ~3 ka does coincide with insolation maximum. Kele et al. [77] find a similarly poor correlation between the timing of tufa formation in the Kurkur-Dungal area (southern Egypt) and northern hemisphere insolation, which is thought to modulate the position of the ITCZ, i.e., high summer insolation is expected to correspond to increased precipitation. However, the southern Egypt tufa deposits do not align with high summer insolation, even when a delay in aquifer recharge is taken into account. They propose that the mechanisms driving rainfall in the region are complex and cannot be attributed to a single forcing factor, e.g. the precession-controlled motion of the ITCZ, and instead conclude that different forcing mechanisms are likely at play during interglacial vs. glacial periods. A potential explanation for the lack of correlation between our Ga-Mohana tufa record and high austral summer insolation is that direct insolation forcing has played a lesser role over the last ~50 ka due to lower amplitude changes related to declining eccentricity [81], and that after ~70 ka, high latitude changes may have had a greater influence on southern African hydroclimate [82]. This could mean that the first and last intervals of tufa formation (~114–100 ka and ~3 ka) may have been driven by low latitude mechanisms, i.e. high austral summer insolation, whereas favourable conditions for tufa formation across MIS 4 and MIS 3, which occurred across variable insolation, may have been a response to high latitude mechanisms, i.e. an increase in the global ice volume.

Following that warmer sea surface temperatures (SST) in the southwest Indian Ocean generate increased moisture and correlate to periods of greater rainfall in southeastern Africa today [79, 83, 84], one might predict that past periods of warmer SST would correspond to periods of tufa formation at Ga-Mohana. However, tufa formation occurs across a range of Indian Ocean SST [62] (Fig 3) suggesting that SST is not the driving mechanism for increased rainfall in this region. While warmer SSTs, coupled with a negative Southern Oscillation Index, is used to explain higher rainfall during the 114–100 ka interval [27], Caley et al. [62] argue that land-sea temperature gradients, rather than SSTs alone, are likely to have played an important role in modulating rainfall variability in southeastern Africa. However, with an ever-growing body of palaeoproxy data generated for sites across southern Africa, it is increasingly acknowledged that broad theoretical models evoking a primary driving mechanism do not capture the homogeneity of rainfall across the region which has experienced a spatially complex pattern of past hydroclimate variability [81, 85]. Through HadCM3 climate model simulations, Singarayer et al. [81] show that correlations between SSTs and rainfall varies spatially and temporally over southern Africa depending on the interplay of different mechanisms. This illustrates that rainfall across the region is driven by the interaction of a range of drivers that wax and wane in prominence, and suggests that the governing mechanism of rainfall variability for any particular region has likely varied over time [81].

It seems clear that tufa formation cannot be simply related to changes in global climate, as these deposits occur in a variety of settings and form in response to variable climatic conditions. However, tufas remain a valuable tool for assessing rainfall regime shifts, karst recharge processes, and changes in environmental parameters, as documented for several deposits world-wide [40, 55, 76, 78, 86–88] and may be complimentary archives to other proxies such as lake sediments and speleothem deposits [39, 60, 89].

The Ga-Mohana tufas form from emergent groundwaters. Recharge of the below surface aquifers, which feed these underground waters, is driven by either increased rainfall, less seasonal / more prolonged rainfall, or reduced effects of evaporation, or most likely a combination of all three. Little is known about the karst dynamics at this site, and so potential lags between rainfall events, aquifer recharge, residence times and tufa formation are as yet unconstrained. However, Markowska et al. [90] show that cooler temperatures and reduced rates of evaporation create favourable conditions for karst recharge in semi-arid regions. As such, tufa deposits at Ga-Mohana Hill are not simply indicators of 'wet' conditions but rather indicate a positive moisture balance caused by an interplay of local and regional climatic and environmental parameters. The time intervals presented here are thus interpreted as periods of prolonged water availability combined with milder temperatures and reduced rates of evaporation. These conditions were met during discrete time periods over the last ~110 ka, the timing of which arise from the interaction of multiple forcing factors.

## Comparison to regional records

We compare the record of tufa formation intervals at Ga-Mohana Hill with other palaeoenvironmental records at nearby Kathu Pan and Wonderwerk Cave (Fig 4). These three sites all occur within ~60 km of each other and thus experienced comparable shifts in local hydroclimate. The tufa record at Ga-Mohana Hill indicates wet conditions during MIS 5d and MIS 4. Sediment analysis at Kathu Pan is consistent with our record; marshy conditions prevailed at Kathu Pan from ~101–80 ka, and palygorskite-coated sands indicate the presence of fluctuating water levels across five intervals between ~167–52 ka [25]. This confirms that the region was wetter during much of MIS 5 and 4. A gap in tufa formation at Ga-Mohana Hill after ~31 ka indicates less water availability during much of MIS 2. This is reflected in the development of extensive pedogenic carbonate deposits at Kathu Pan by ~23 ka, which indicate drier conditions and perhaps more seasonal rainfall compared to earlier time periods [25]. At Wonderwerk Cave, a hiatus in stalagmite growth after ~33 ka [19] is consistent with drier conditions, however, wetter conditions from ~23 to 17 ka are reflected in the pollen and stable isotope record from the same stalagmite; this evidence for wetter conditions at Wonderwerk Cave during the LGM is inconsistent with the records from Ga-Mohana and Kathu Pan. Ga-Mohana Hill documents a subsequent late glacial wet period commencing as early as ~15 ka, while at Wonderwerk Cave, slow growth of the stalagmite between ~17–13 ka signifies reduced moisture availability [19], and pedogenic carbonates at Kathu Pan, indicating dry conditions, persist at ~10 ka [25]. The most recent period of tufa formation at Ga-Mohana is at ~3 ka. Through the Holocene, fine layers of organic material alternate with calcium carbonate deposits at Kathu Pan, implying an increased amplitude of fluctuating water availability and aridity [25], and at Wonderwerk Cave, stalagmite growth resumes at ~3.5 ka, indicating wet conditions during the late Holocene [18, 19]. The earlier and latest parts of the sequences from these sites are consistent, but the record from Wonderwerk Cave indicating wetter conditions during MIS 2 is inconsistent, and some discrepancy between the records exist for the MIS 2—Holocene transition.

The close proximity of Ga-Mohana Hill, Kathu Pan, and Wonderwerk Cave to each other makes it possible that they were utilized by the same groups of mobile hunter-gatherers, thus providing an opportunity to consider the relationships between wet periods and evidence for human occupation in this region of the southern Kalahari (Fig 4). Between ~251 and 138 ka at Wonderwerk Cave, there is evidence for both wetter conditions and human occupation [20]. Archaeological material at Kathu Pan occurs within palygorskite-coated, water-associated sediments dated to ~156 ka, ~121 ka and ~74 ka [25], with the latter being a Howiesons Poort

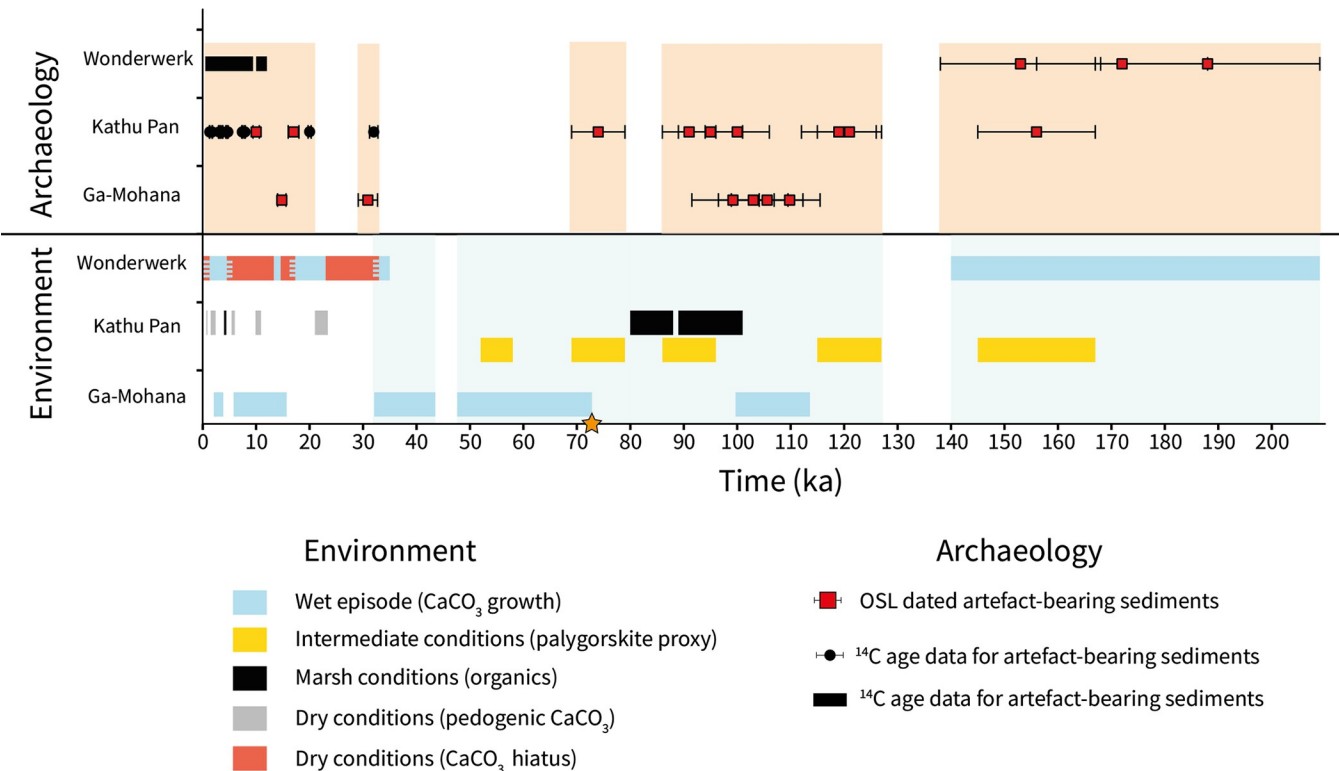

**Fig 4. Comparison of palaeoenvironmental and archaeological records from Ga-Mohana Hill, Kathu Pan, and Wonderwerk Cave.** Archaeological occupation ages for Ga-Mohana Hill [27, 36], Kathu Pan [25, 26, 91, 92] and Wonderwerk Cave [20, 93, 94]. Pale orange bars highlight periods of occupation. Palaeoenvironment proxy data from Ga-Mohana Hill (this study), Kathu Pan [25] and Wonderwerk Cave [19, 20]. Pale blue bars highlight wet periods across the sites. Orange star marks the point at ~71 ka, before which human occupation of the region appears to have been associated with the availability of water.

occurrence. Also at Kathu Pan, wet, marshy conditions that are likely to have supported a significant amount of vegetation occur between ~101 and 80 ka, coupled with evidence for human occupation [25]. At Ga-Mohana Hill, tufa formation at ~114–100 ka correlates with human occupation at the site. Thus, in summary, before ~71 ka, human occupation of the region appears to have been associated with the availability of water.

After ~71 ka, the timing of human occupation and wet periods do not coincide (Fig 4). Tufas at Ga-Mohana Hill indicate that much of MIS 4 and 3 is characterised by wet conditions. The sediments at Kathu Pan continue to indicate the presence of water through much of MIS 4, although the organic-rich marsh sediments do not occur after MIS 5 [25]. However, evidence for human occupation during this time is lacking at both Kathu Pan and Ga-Mohana Hill [36]. There are Middle Stone Age deposits at Wonderwerk Cave that have not yet been securely dated that could potentially represent this period [9], but this remains unknown at this point. From ~43–32 ka, wet conditions are represented at Ga-Mohana Hill, and from ~35–33 ka at Wonderwerk Cave [19]. In contrast, there is evidence for drier conditions from ~32 ka at Kathu Pan [25]. Evidence for human occupation at Ga-Mohana Hill [36] and Kathu Pan [92] date to 31 ± 1.8 ka and 32 ± 0.78 ka, respectively. These data suggest that human occupation at these sites may overlap with a period of decreasing water availability, but the error ranges on the age estimates make it challenging to confidently assert this.

Human occupation is evident again at Ga-Mohana Hill at 14.8 ± 0.8 ka [36] and is associated with evidence for relatively wetter conditions, but at Kathu Pan, human occupation during the LGM [91, 92] is associated with evidence for relatively drier conditions. Late glacial

deposits at Wonderwerk Cave indicate an association of dry conditions and human occupation [94]. Thus, MIS 2 provides very little coherence with respect to the relationship between water availability and human occupation; humans are associated with both wet and dry conditions. Through the Holocene, there is persistent evidence for human occupation despite changes in palaeoenvironmental conditions [25, 92, 94].

## Discussion

In this semi-arid region with limited, seasonal rainfall and no evidence of actively precipitating tufa, the relict tufa deposits at Ga-Mohana Hill are a record of past periods of conditions favourable for tufa formation, which are primarily an indication of increased water on the landscape. We show that U-Th dating of the tufas, buoyed by the laser ablation screening method, can produce precise ages. We go on to use these ages to show that periods of tufa formation were punctuated over the last 110 ka, with five discrete time periods identified.

Our U-Th dated tufa records suggests periods of increased water availability in the southern Kalahari were not restricted to interglacials. Ga-Mohana Hill shows extensive tufa formation during much of MIS 4, a period generally assumed to be characterised by typical cold and dry glacial conditions across much of the interior of southern Africa [95]. Increased water availability during this time is supported by other palaeoenvironmental records of the Kalahari Basin, such as at Kathu Pan. At Witpan Dunes, approximately 350 km to the north west, the absence of southern Kalahari dune data during MIS 4 [96] indicates unfavourable conditions for dune accumulation, suggesting increased rainfall, decreased windiness, and a denser vegetation cover [97]. To the north, a Makgadikgadi Megalake highstand has been dated to 64.2 ± 2.0 ka suggesting there was also substantial water availability in the Middle Kalahari at that time [98]. We argue that the tufa intervals represent a southern Kalahari environment characterised by a positive hydrological balance and mild temperatures favourable for productive vegetation and soils. In accordance with other recent studies, our results challenge global generalisations of past climate change and highlight the necessity for regionally specific models [16, 21, 85].

In the southern Kalahari, early human population distributions appear to have been modulated by water availability before ~71 ka. After ~71 ka, the picture is much less clear. Despite evidence for wetter conditions, archaeological deposits dating to MIS 4 and the early part of MIS 3 have not yet been identified in the punctuated record of human occupation at Ga-Mohana Hill, nor at nearby Kathu Pan or Wonderwerk Cave. This result poses a new dilemma, in that wet conditions in this region of the southern Kalahari should have theoretically made it attractive for human occupation, but as of yet, no archaeological deposits date to this time. The time interval corresponding to MIS 2 provides little coherence with respect to the relationship between water availability and human occupation. The three records considered here do not agree on whether conditions were wetter or drier during the LGM and humans appeared to have occupied the region through both the LGM and late glacial. Others have highlighted that the palaeoenvironmental record for MIS 2 across the Kalahari Basin and surrounding regions is complex, documenting a high degree of spatial and temporal variability [16]. This lack of coherence may be in part due to the variable responses of palaeoenvironmental proxies to temperature and water availability changes, and potentially lags in responses. A shift in seasonality may also play a role, with some proxies responding to seasonality changes for precipitation, as evidenced at Kathu Pan [25], as opposed to mean annual precipitation.

We note that the absence of evidence may not be evidence of absence, and that issues with site formation, site visibility, and/or dating may instead explain why no archaeological deposits have yet been identified. The inability to fully explain this pattern is one of this study's

limitations. Further work, including the excavation and dating of new archaeological sites, will be key for further testing hypotheses that link early human population distribution patterns to water availability, potential refugia conditions, and interglacial/glacial cycling [4–6, 8, 9].

## Conclusion

We have identified and described the tufa deposits at Ga-Mohana Hill, and provided a framework for reliably dating them. The observed tufa morphologies suggest a depositional environment characterised by water cascading down the dolomite steps, flowing over the rock shelters, and ponding in shallow pools. Through U-Th dating, we have produced a new, well-dated record of prolonged water availability linked to human occupation in the southern Kalahari during the Late Pleistocene. Identifying the timing and nature of human occupation in the Kalahari Desert is critical for understanding the emergence of our ability to adapt to new and extreme environments [2]. For a long time, the Kalahari Desert has been considered too arid for early human populations to persist, and evidence for occupation was assumed to represent wetter periods. Until now, a rarity of integrated palaeoenvironmental and archaeological records has largely prevented adequate testing of these assumptions. The results presented here provide evidence for prolonged periods of water availability in the southern Kalahari Basin through much of the Late Pleistocene. There is a positive association with wet conditions and human occupation before ~71 ka. However, by the LGM, water availability alone did not mediate human occupation in the southern Kalahari Desert. This may extend further back in time as the datasets for the time period between ~71 ka and the LGM become more robust. Nevertheless, this result challenges the traditional view that links wet periods to human occupation. This decoupling of human occupation and wet phases in the Late Pleistocene could reflect new social and technological adaptations that helped hunter-gatherers cope more effectively with diverse environmental conditions.

## Supporting information

**S1 Checklist. *PLOS ONE* inclusivity in global research questionnaire.**
(PDF)

**S1 Fig. Photographs of cascade hand and drill core samples.** a,b) field context of sample 18–4; c) hand sample scan of 18–4 showing fine, undulating layers; d,e) field context of drill core samples 18–17 (f) and 18–16 (g); h) photograph of in-situ cascade tufa sampled with core drill; i) drill core sample 18–12; j,k) thin section photographs of sample 18–12 (in ppl) showing irregular, domal micritic laminae with lenses of microspar and micropores.
(PDF)

**S2 Fig. Field photographs, hand samples and thin section photomicrographs of rim pool samples.** a) Field context of sample 17–16 showing circular configuration and surface desiccation cracks; b,c) hand sample photographs showing 4cm layer of carbonate and cemented clasts on the underside; d) field context of sample GHN-1; e) hand sample scan of GHN-1; c) photograph of thin section from GHN-1 showing aphanitic fabric of biomicrite with spar-filled filamentous cavities.
(PDF)

**S3 Fig. Photographs of terrace breccia hand samples.** a) terrace breccia sample showing included detrital clasts and brecciated tufa clasts; b) hand sample scan of sample 18–7; c) field context and d) hand sample scan of sample 17–8 showing massive, dense micrite; e,f) thin section photographs of terrace sample 17–8 showing clotted fabric of peloidal micrite with

microspar-filled void spaces.
(PDF)

**S4 Fig. Photographs, hand sample scans and thin section photographs of barrage tufas.** a) field photograph of sample 17–6; b) hand sample scan of sample 17–6 showing irregular, undulating and discontinuous layering; c) thin section photographs show stromatolite-type micrite crustal laminae alternating with chaotic microspar laminae with detrital and oxide inclusions (c) and discontinuous crinkly microspar laminae with overprinting of oxide precipitates (f).
(PDF)

**S5 Fig. Photographs of sampled domes.** a) field context of tufa dome, sampled using an angle grinder; b) dense mm-scale layers alternating with irregular, porous and friable layers; c) photomicrograph of thin section from sample in (b) showing micro laminae; d,e) dome sampled with drill-core; f) hand sample of dome core showing large cavities and porous, reticulate framework.
(PDF)

**S6 Fig. High resolution images of cascade (GHN2 and GHS5), terrace (17–8) and rim pool (GHN1, 18–7, GHS6) tufa samples overlain by LA-ICP-MS $^{238}$U (left) and $^{232}$Th (right) element distribution maps.** Concentrations in ppm shown in adjacent colour scales (warmer colour = higher concentration). Black circles represent approximate locations of subsamples drilled for U-Th dating prior to pre-screening, and oblong free-forms show exact locations at which subsamples were drilled for U-Th dating following LA-ICP-MS analysis. Ages associated with each subsample are given in thousands of years (ka) and are reported in Table 1.
(PDF)

**S7 Fig. High resolution images of cascade core samples (18–13, 18–14, 18–15, 18–16, 18–17) overlain by LA-ICP-MS $^{238}$U (left) and $^{232}$Th (right) element distribution maps.** Concentrations shown in ppm in adjacent colour scales (warmer colour = higher concentration). Black circles represent approximate locations of subsamples drilled for U-Th dating prior to pre-screening, and oblong free-forms show exact locations at which subsamples were drilled for U-Th dating following LA-ICP-MS analysis. Ages associated with each subsample are given in thousands of years (ka) and are reported in Table 1.
(PDF)

**S8 Fig. High resolution images of samples with unreliable and imprecise age solutions (S3 Table) with the exception of sample 18–10.2 (bottom of sample) which has an age of 3.0 ± 0.9 ka.** Black oblong outline represents material drilled for U-Th dating. Cascade samples (18–4 and 18–12), barrage samples (17–6 and 18–6) and dome core samples (18–10) overlain by LA-ICPMS $^{238}$U (left) and $^{232}$Th (right) element distribution maps. Concentrations in ppm shown in adjacent colour scales (warmer colour = higher concentration).
(PDF)

**S1 Table. Tufa sample inventory. Samples labelled and listed in order of collection.** Samples with GH prefix collected in 2016, numerical prefix of other samples indicates the year they were collected (e.g. 17- = 2017).
(PDF)

**S2 Table. U and Th isotope ratios measured in tufa samples with reliable and precise ages.** The samples are labelled according to the sequence they were collected in but presented in stratigraphic order. Errors on all isotope activity ratios are reported with 2σ uncertainty. All

ages have been corrected to account for the effect of detrital Th assuming an estimate for initial $^{230}$Th/$^{232}$Th of 1.5 ± 1.5, and calculated using the $^{230}$Th-$^{238}$U decay constants of Cheng et al. [54] and equation 1 from Hellstrom [47].
(PDF)

**S3 Table. U and Th isotope ratios measured in tufa samples which have unreliable or imprecise age solutions.** Errors on all isotope activity ratios are reported with 2σ uncertainty. Upper limit is defined as corrected age plus 2σ uncertainty. All ages have been corrected to account for the effect of detrital Th assuming an estimate for initial $^{230}$Th/$^{232}$Th of 1.5 ± 1.5, and calculated using the $^{230}$Th-$^{238}$U decay constants of Cheng et al. [54] and equation 1 from Hellstrom [47].
(PDF)

## Acknowledgments

Thank you to the Baga Motlhware Traditional Council and South African Heritage Resources Agency for permissions to work at Ga-Mohana Hill. We acknowledge Serene Paul, Roland Maas, Bence Paul and Russell Drysdale at the University of Melbourne for assistance with the U-Th analyses and laser ablation, and Renee Van Der Merwe at the University of Cape Town for preparing thin sections. We also acknowledge David Morris and the McGregor Museum, Simon Hall, Andy Herries, Kyle Brown and Sechaba Maape.

## Author Contributions

**Conceptualization:** Jessica von der Meden, Robyn Pickering, Benjamin J. Schoville, Jayne Wilkins.

**Data curation:** Jessica von der Meden.

**Formal analysis:** Jessica von der Meden, Helen Green, Rieneke Weij, John Hellstrom, Alan Greig.

**Funding acquisition:** Robyn Pickering, Benjamin J. Schoville, Jayne Wilkins.

**Investigation:** Jessica von der Meden.

**Project administration:** Jessica von der Meden, Robyn Pickering, Jayne Wilkins.

**Resources:** Jon Woodhead.

**Supervision:** Robyn Pickering, Jayne Wilkins.

**Visualization:** Jessica von der Meden, Benjamin J. Schoville.

**Writing – original draft:** Jessica von der Meden.

**Writing – review & editing:** Jessica von der Meden, Robyn Pickering, Benjamin J. Schoville, Helen Green, Rieneke Weij, John Hellstrom, Jon Woodhead, Wendy Khumalo, Jayne Wilkins.

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
