## [Decision Letter · Decision Letter 0]

7 Mar 2022

PONE-D-22-02819Tufas indicate decoupling of water availability and human occupation in the southern KalahariPLOS ONE

Dear Dr. von der Meden,

Thank you for submitting your manuscript to PLOS ONE. After careful consideration, we feel that it has merit but does not fully meet PLOS ONE’s publication criteria as it currently stands. Therefore, we invite you to submit a revised version of the manuscript that addresses the points raised during the review process.

The two reviewers greatly appreciated your work and conclusions and I also agree that your paper meets the standard of publication required by PLoS ONE, pending minor comments. Among the suggestions from the two reviewers, I would remark that the tight dichotomy wet/dry (interglacial/glacial) in the formation of spring tufa (and also speleothems) in arid regions is simplistic and many studies are confirming this. In my experience in the hyperacid Sahara and arid Dhofar Mountains of Oman, I noticed that even today spring tufa form, evidently with different sedimentary patterns respect to when they formed during pluvial phase. I would appreciate (as suggested by Rev2) a deeper discussion on this topic, confirming that dealing with spring tufa requires to do beyond the equation wet->deposit, dry->no deposition.

We look forward to receiving your revised manuscript.

Kind regards,

Andrea Zerboni, Ph.D.

Academic Editor

PLOS ONE

Journal Requirements:

2. In your Methods section, please provide additional information regarding the permits you obtained to collect samples for the present study. Please ensure you have included the full name of the authority that approved the field site access and, if no permits were required, a brief statement explaining why.

3. Please include a complete copy of PLOS’ questionnaire on inclusivity in global research in your revised manuscript. Our policy for research in this area aims to improve transparency in the reporting of research performed outside of researchers’ own country or community. The policy applies to researchers who have travelled to a different country to conduct research, research with Indigenous populations or their lands, and research on cultural artefacts. We note that some of the authors of your manuscript are from outside of South Africa. The questionnaire can also be requested at the journal’s discretion for any other submissions, even if these conditions are not met.  Please find more information on the policy and a link to download a blank copy of the questionnaire here: https://journals.plos.org/plosone/s/best-practices-in-research-reporting. Please upload a completed version of your questionnaire as Supporting Information when you resubmit your manuscript.

4. We note that Figure 1 in your submission contain copyrighted images. All PLOS content is published under the Creative Commons Attribution License (CC BY 4.0), which means that the manuscript, images, and Supporting Information files will be freely available online, and any third party is permitted to access, download, copy, distribute, and use these materials in any way, even commercially, with proper attribution. For more information, see our copyright guidelines: http://journals.plos.org/plosone/s/licenses-and-copyright.

a) You may seek permission from the original copyright holder of Figure 1 to publish the content specifically under the CC BY 4.0 license. 

Additional Editor Comments (if provided):

Reviewers' comments:

Reviewer's Responses to Questions

**Comments to the Author**

1. Is the manuscript technically sound, and do the data support the conclusions?

Reviewer #1: Partly

Reviewer #2: Yes

2. Has the statistical analysis been performed appropriately and rigorously? 

Reviewer #1: Yes

Reviewer #2: N/A

3. Have the authors made all data underlying the findings in their manuscript fully available?

Reviewer #1: Yes

Reviewer #2: Yes

4. Is the manuscript presented in an intelligible fashion and written in standard English?

Reviewer #1: Yes

Reviewer #2: Yes

5. Review Comments to the Author

Reviewer #1: In overview I think this is a useful paper. The substantive part of the paper is about dating tufas using sensible new methods to choose the best samples for U-Series. My suggestions are therefore relatively minor, in terms of the interpretation. This is a question of minor revisions in my opinion. The authors do a good job at showing an interesting palaeoenvironmental story in terms of tufas. But I am dubious about what this says about decoupling of humans and water. So I think this aspect of interpretation should be removed, and the paper will be stronger. The key conclusion is actually about humid conditions over a long period in Kalahari than previously known.

Lines 40/45 – all very southern African references. Not surprising given the study area and the authors – but I think a bit of extra punch here would be given by first looking more widely at Africa before zooming in on the Kalahari. E.g. Basell (2008, QSR) on East Africa…Scerri et al. (2014, QSR) on North Africa etc. For instance, in the former, Basell argues that in East Africa humans are seemingly tethered to water sources until MIS 5, and then seemingly not. Seems relevant to me. So I suggest just an initial paragraph on such stuff. It I also useful to think about some of complexities here…e.g. in North Africa most human occupations correlate with wet episodes…but some seemingly don’t. Uan Tabu, for instance. But this is dated by OSL, done when this was an experimental method, so maybe the dates are wrong. So as well as a bit of a wider geographical focus, a bit of intro on methods stuff would be useful.

I think dating tufas is a great way to go for reconstructing humidity in the study area, just nice to situate his more widely in terms of different archives and the history of study.

HG – to think about. Sampling intensity, at one point are enough samples to reliably reconstruct human occupations and regional climate present

Using the LC-ICP-MS method seems exciting. The methods are described very clearly and all seem sensible, i.e. locating areas with less detrital contamination for dating. This sounds like a method which could be used more widely in tufa studies in the future. I am not an expert at U-Series dating so I cannot comment on specific technical aspects, but I could follow the authors’ description, it all makes sense, and seems reasonable. Likewise, the geological description of the study area is well written. I hope that a U-Series expert also reviews this paper to comment on this aspect.

The typical problem with studies like this is that there will be one or two age estimates for humidity in an area, and it is hard to know how representative they are. The results here show clusters of ages, suggesting wet phases ca. 114-100 ka, 73-48 ka, 44-32 ka and 15-2 ka….this mostly seems fine. However, I am not totally sure about the separation of the middle two groups. Out of interested I plotted the dates (rounded slightly, not considering uncertainty) dating to MIS 4 and 3 out of interest, and it looks like a continuous series to me, not one that is two groups. I’m dubious that this self-evidently represents two distinct humid phases….another way of looking at this…is that the ca. 48 ka age is as close to the oldest date in the 44-32 ka group as it is to the next youngest date in the 73-48 ka group….So I suggest the authors clarify their thinking a little here. To me, a simpler interpretation is that there are three clusters of ages, 114-100 ka, 73-32 ka, and 15-2 ka. I note the authors themselves suggest “our record suggests tufa formation was semi-continuous across MIS 4 and MIS 3” (line 278).

It might also be worth thinking about sampling adequacy and how meaningful the inferred gaps are.

In terms of interpretation. They describe how earlier phases of occupation correlate with wet phases, but that there is a decoupling of this after 71 ka. But I don’t really understand this, as their figure 4 suggests that no human occupations are known in the area between the end of MIS 5 and the end of MIS 3. Surely the point about humans not being tethered to water would be shown by human occupations during arid times? Their argument is that they have shown there wet condition, but no evidence for people means this de-tethering. I get what they mean, but not sure how strong an argument that is. I think a simpler interpretation is that with so few archaeological sites in the area dated, it is simply a question of sampling adequacy. If it was really so wet, why would humans not have been in the area? And they even point to possible evidence at Wonderwerk which could date to this MIS 4/3 time. It is a question of sampling adequacy. Do three sites really demonstrate the absence of people from the region? If we only had Wonderwerk and Ga-Mohana, it would appear no people were in the area between ca. 90 and 30 ka. Yet Kathu Pan shows people there at about 75 ka. Who is to say another site would not show people at 65 ka, 55 ka, etc?

It is with the end of MIS 3 and into MIS 2 that more persuasive evidence for decoupling of humans and water is evident, with humans being found in seemingly arid situations. With this evidence it could be stated that this decoupling seemed to happen around 30 ka.

Even if their ‘decoupling’ argument is correct, it does not really make sense. If humans were living in arid areas this would indeed say something about “new social and technological adaptations that helped hunter-gatherers cope more effectively with diverse environmental conditions” (line 426/427). But the absence of people from what are supposedly good conditions begs the question ‘why was no one there?’ rather than suggesting that some kind of adaptations are involved in people not being somewhere nice.

In summary: this is a useful paper, but I suggest the authors change their interpretation somewhat.

Reviewer #2: The study of von der Maden et al. concerns the study of fossil tufa deposits form the arid region of southern Kalahari in unraveling their connection with local archeological human origins and the impact for the palaeoenvironmental reconstructions.

I have appreciated the study, the obtained ages and the resulting data concerning implications with human settling

Many studies emphasize the close relation.

My only remarks are about:

Tufa morphological components (line 162): there are several morphological classifications about tufa depositional system. As done for the fabric classification (Manzo et al.), I suggest indicating the reference used for the morphological classification (definitions are available from Pentecost and Viles, 1994, Pedley 2009; Arenas-Abad et al. 2010; Jones and Renaut 2010, Capezzuoli et al., 2014),

Comparison to global records (line 271-311): in my opinion this is paragraph is a bit ambitious, not only because the herereported bibliography is really poor (please note that there is already a review article on tufa/travertine deposition and climate…and it is not cited: Ricketts, J.W., Ma, L., Wagler, A.E. and Garcia, V.H. 2019. Global travertine deposition modulated by oscillations in climate. Journal of Quaternary Science, 34, 558–568, https://doi.org/10.1002/jqs.3144), but because the tufa deposition is really different in different setting/climatic regimes of the world. I agree with authors that the dichotomy wet/dry – interglacial/glacial view is to be considered simplistic (Line 275-276), but their consequent suggestion that tufa formation is not restricted to interglacial periods is not new!

For review between climate and tufa deposition, I suggest to see Andrews, 2006; Pedley, 2009, while about the former presence of active tufas as record of important rainfall regime shifts, this has been already showed in distal glacial (South Europe, Capezzuoli et al., 2010; Alexandrowicz, 2012), in semi-arid environments (Brazil, Auler & Smart, 2001; Spain, Luzon et al., 2011) and desert settings (Namibia, Viles et al., 2007; Libya, Cremaschi et al., 2010; Ethiopia, Moeyersons et al., 2006). In contrast, the presence of tufas deposits in tropical and monsoon-dominated settings testifies to an absence of destructive large wet season floods and, consequently, for reduced periods of rainfall (Carthew et al., 2003, 2006).

In the correct hypothesis to study the local weather conditions (rainfall and insolation) as constraining for the local tufa deposition, I suggest to also consider and compare with the Egyptian tufa described in Kele et al. 2021 (https://doi.org/10.1144/jgs2020-147) and relative discussion implementing insolation, humidity, radiometric dating and isotope (also showing that the record of climate in Egypt’s tufa is inconsistent with a simple model of palaeoclimate for this region).

Very minor trifles are indicated in the attached pdf

Hope it helps

Enrico Capezzuoli

6. PLOS authors have the option to publish the peer review history of their article (what does this mean?). If published, this will include your full peer review and any attached files.

Reviewer #1: No

Reviewer #2: **Yes: **Enrico Capezzuoli

---

## [Author Response · Author response to Decision Letter 0]

18 May 2022

Manuscript PONE-D-22-02819

Response to Reviewers

Dear Professor Zerboni,

Thank you for the opportunity to submit a revised draft of our manuscript Tufas indicate prolonged water availability linked to human occupation in the southern Kalahari (reference PONE-D-22-02819) for publication in the journal PLOS ONE. We appreciate the time and effort that you and the reviewers put in to providing useful feedback on our manuscript and we are grateful for the insightful comments and suggestions which definitely improve our paper. 

The main issues raised by the reviewers are highlighted below, along with a summary of how we addressed them.

1) Our interpretation of the combined palaeoenvironment and archaeological records as indicating a decoupled relationship between water availability and human occupation at ~71 ka. We addressed this by clarifying and strengthening our interpretation about the relationship between water availability and human occupation. We highlight evidence that water availability and human occupation are indeed correlated until 71 ka. By ~20 ka during the LGM, humans were occupying this region of the southern Kalahari despite evidence for variable (drier) climatic conditions, as opposed to earlier times when occupation appears only during periods of increased water availability (as identified by our tufa ages). We have changed the title so that there is less emphasis on the decoupling and instead highlight the new 110,000 year long tufa record of water availability in the southern Kalahari.

2) Limitations in terms of sampling adequacy. We addressed this by making clear the limitations of this study by acknowledging the challenge to explain gaps in the archaeological record during periods when the palaeoclimate data suggests favourable conditions for human activity in the region. We point to issues with site formation, visibility and dating as potential reasons this gap persists, and highlight the need for newly excavated and well-dated sites to resolve these issues.

3) A request for an acknowledgement that the association of tufas with interglacial climate conditions is an outdated one, and a deeper discussion on the nuanced climatic conditions that tufas represent. We addressed this by including two new paragraphs to discuss the nuance around the presence of tufa deposits, and that they do not simply equate to wet conditions. Instead we argue that tufas form during both glacial and interglacial periods (citing previous studies), and in response to a complex interplay of a variety of environmental parameters, including reduced rates of evaporation and cooler temperatures. 

4) The inclusion of several additional references to provide more context and to acknowledge new and relevant studies. We have addressed this by incorporating the suggested references and relevant information from the respective studies. We have thus updated our reference list. 

We respond to each of the reviewers’ comments in red below. The changes are tracked within the manuscript and the line numbers referenced below refer to the revised manuscript with tracked changes. 

Editor

The two reviewers greatly appreciated your work and conclusions and I also agree that your paper meets the standard of publication required by PLoS ONE, pending minor comments. 

Thank you!

Among the suggestions from the two reviewers, I would remark that the tight dichotomy wet/dry (interglacial/glacial) in the formation of spring tufa (and also speleothems) in arid regions is simplistic and many studies are confirming this. In my experience in the hyperacid Sahara and arid Dhofar Mountains of Oman, I noticed that even today spring tufa form, evidently with different sedimentary patterns respect to when they formed during pluvial phase. I would appreciate (as suggested by Rev2) a deeper discussion on this topic, confirming that dealing with spring tufa requires to do beyond the equation wet->deposit, dry->no deposition.

We thank the editor for this comment, and trust that the additions made to the text (detailed in response to the comments below) sufficiently address the complexity of tufa deposits and the environmental conditions they represent. We have added several statements to the text in the “Comparison to global records” section (lines 343-438) to acknowledge that tufas form in response to a variety of climatic factors, and do not merely represent wet / dry conditions (e.g. 353-357; 430-434). In particular, we have included an additional paragraph in which we discuss the nuanced conditions required for tufa formation at Ga-Mohana Hill (lines 426-438). We have also adjusted the language we use throughout the text to move away from the simplistic implications of the term ‘wet’ when referring to the conditions that tufas represent (e.g. lines 127-129).

Reviewer #1

In overview I think this is a useful paper.

Thank you!

The substantive part of the paper is about dating tufas using sensible new methods to choose the best samples for U-Series. My suggestions are therefore relatively minor, in terms of the interpretation. This is a question of minor revisions in my opinion. The authors do a good job at showing an interesting palaeoenvironmental story in terms of tufas. But I am dubious about what this says about decoupling of humans and water. So I think this aspect of interpretation should be removed, and the paper will be stronger. The key conclusion is actually about humid conditions over a long period in Kalahari than previously known.

Thank you. We have put more emphasis on the evidence for water availability as represented by the tufas . We have clarified that we have good evidence for correlation between wet periods and human occupation before 71 ka, and a decoupling of this relationship by the LGM. There are more specific comments about this interpretation below where the Reviewer expands on their point.

Lines 40/45 – all very southern African references. Not surprising given the study area and the authors – but I think a bit of extra punch here would be given by first looking more widely at Africa before zooming in on the Kalahari. E.g. Basell (2008, QSR) on East Africa…Scerri et al. (2014, QSR) on North Africa etc. For instance, in the former, Basell argues that in East Africa humans are seemingly tethered to water sources until MIS 5, and then seemingly not. Seems relevant to me. So I suggest just an initial paragraph on such stuff. 

It I also useful to think about some of complexities here…e.g. in North Africa most human occupations correlate with wet episodes…but some seemingly don’t. Uan Tabu, for instance. But this is dated by OSL, done when this was an experimental method, so maybe the dates are wrong. So as well as a bit of a wider geographical focus, a bit of intro on methods stuff would be useful.

Thank you for the reference suggestions and for raising these points. We have added a paragraph to the Introduction (lines 46-56) to provide additional context from studies in other parts of Africa. We have summarised the key conclusions presented in the suggested studies which address the relationship between water availability and Homo sapiens’ mobility in eastern and northern Africa. We have also included an introductory sentence on the methods used to investigate questions relating to human-environment dynamics (lines 56-59).

I think dating tufas is a great way to go for reconstructing humidity in the study area, just nice to situate his more widely in terms of different archives and the history of study.

HG – to think about. Sampling intensity, at one point are enough samples to reliably reconstruct human occupations and regional climate present

Thank you for this suggestion. Although we agree that it is an important consideration, restrictions relating to funding to support field work and sample analysis means that we are limited to work with what we have got so far. In terms of the number of dated tufa samples, we were constrained by the quality of material that can provide a reliable age; in total over 40 tufa samples have been collected, but only 16 of these samples met the criteria for U-Th age dating. Similarly, the number of excavated archaeological sites that date to each time period are limited. Nonetheless, we believe that these data are valuable and do shed light on patterns of the past, even if imperfectly. We have made clearer in the Discussion section what the study’s limitations are, i.e. the challenge to explain gaps in the archaeological record during periods when the palaeoclimate data suggests favourable conditions for human activity in the region (lines 558-560 and 572-575). We point to issues with site formation, visibility and dating as potential reasons this gap persists.

Using the LC-ICP-MS method seems exciting. The methods are described very clearly and all seem sensible, i.e. locating areas with less detrital contamination for dating. This sounds like a method which could be used more widely in tufa studies in the future. I am not an expert at U-Series dating so I cannot comment on specific technical aspects, but I could follow the authors’ description, it all makes sense, and seems reasonable. Likewise, the geological description of the study area is well written. I hope that a U-Series expert also reviews this paper to comment on this aspect.

Thank you.

The typical problem with studies like this is that there will be one or two age estimates for humidity in an area, and it is hard to know how representative they are. The results here show clusters of ages, suggesting wet phases ca. 114-100 ka, 73-48 ka, 44-32 ka and 15-2 ka….this mostly seems fine. However, I am not totally sure about the separation of the middle two groups. Out of interested I plotted the dates (rounded slightly, not considering uncertainty) dating to MIS 4 and 3 out of interest, and it looks like a continuous series to me, not one that is two groups. I’m dubious that this self-evidently represents two distinct humid phases….another way of looking at this…is that the ca. 48 ka age is as close to the oldest date in the 44-32 ka group as it is to the next youngest date in the 73-48 ka group….So I suggest the authors clarify their thinking a little here. To me, a simpler interpretation is that there are three clusters of ages, 114-100 ka, 73-32 ka, and 15-2 ka. I note the authors themselves suggest “our record suggests tufa formation was semi-continuous across MIS 4 and MIS 3” (line 278). It might also be worth thinking about sampling adequacy and how meaningful the inferred gaps are.

Thank you for raising this point. We define the wet episodes based on clusters of overlapping tufa ages, with the bounds defined by the minimum / maximum age calculated using the uncertainties of the ages. We have added this explanation to the manuscript (lines 319-320), and the following sentence has been added to Figure 3 caption to clarify our thinking as suggested (lines 331-334): “The blue bars highlight clusters of overlapping tufa ages and are defined by the minimum and maximum ages in each range, calculated using the 2σ uncertainty associated with the ages. Based on the presence of the tufa deposits, these periods are inferred to represent episodes of greater water availability on the landscape.” We do not infer wet periods where there is no age data, hence our statement of ‘semi-continuous’ over the MIS 4-3 period. In order to be consistent, we have adjusted the fourth episode of tufa formation to include only the two ages that overlap, and have separated the youngest age from this cluster as there is in fact no overlap. We adjusted the text to account for this fifth episode throughout. 

In terms of interpretation. They describe how earlier phases of occupation correlate with wet phases, but that there is a decoupling of this after 71 ka. But I don’t really understand this, as their figure 4 suggests that no human occupations are known in the area between the end of MIS 5 and the end of MIS 3. Surely the point about humans not being tethered to water would be shown by human occupations during arid times? Their argument is that they have shown there wet condition, but no evidence for people means this de-tethering. I get what they mean, but not sure how strong an argument that is. I think a simpler interpretation is that with so few archaeological sites in the area dated, it is simply a question of sampling adequacy. If it was really so wet, why would humans not have been in the area? And they even point to possible evidence at Wonderwerk which could date to this MIS 4/3 time. It is a question of sampling adequacy. Do three sites really demonstrate the absence of people from the region? If we only had Wonderwerk and Ga-Mohana, it would appear no people were in the area between ca. 90 and 30 ka. Yet Kathu Pan shows people there at about 75 ka. Who is to say another site would not show people at 65 ka, 55 ka, etc?

Yes – this is a good point. We had included a statement about ‘absence of evidence is not necessarily evidence of absence’, and now we have made it even clearer in the discussion that this is one of the study’s limitations. There is no immediate solution to this problem – the only solution being to dig and date more sites. 

That said, the three well-stratified, well-dated sites with overlapping MSA chronologies do provide a lot of value in addressing this environment-human interaction question. There are few situations like this across the African continent. For that reason, we believe these results have a lot of value. It’s true that they do pose a new dilemma – where were people during MIS 4 and the early part of MIS 3 when the southern Kalahari was wetter than it is today? We hope further investigations will help reveal the answer. 

Our study has been able to demonstrate correlation between wet periods and human occupation before 71 ka, and a lack of correlation by the LGM. Humans were able to occupy diverse conditions (wet or dry) in the southern Kalahari by the LGM. Our results also suggest that this capacity may predate the LGM, pending further investigations that improve as the archaeological and palaeoenvironmental records become more robust. 

It is with the end of MIS 3 and into MIS 2 that more persuasive evidence for decoupling of humans and water is evident, with humans being found in seemingly arid situations. With this evidence it could be stated that this decoupling seemed to happen around 30 ka.

Thank you. We have modified the language we use to add clarity, particularly in the Conclusion section. We now say “However, by the LGM, water availability alone did not mediate human occupation in the southern Kalahari Desert. This may extend further back in time as the datasets for the time period between ~71 ka and the LGM become more robust” (lines 608-612). We also discuss the records for human occupation and palaeoclimate at the end of MIS 3 (lines 506-512) and, although it appears that human occupation coincides with the end of a wet period and the onset of drying conditions, the errors on the available data make it difficult to confidently assert whether human occupation actually corresponds with wet or dry phases, or both.

Even if their ‘decoupling’ argument is correct, it does not really make sense. If humans were living in arid areas this would indeed say something about “new social and technological adaptations that helped hunter-gatherers cope more effectively with diverse environmental conditions” (line 426/427). But the absence of people from what are supposedly good conditions begs the question ‘why was no one there?’ rather than suggesting that some kind of adaptations are involved in people not being somewhere nice.

Point taken. Hopefully, the comments above and the edits we have made to the language resolve this concern. This statement about new adaptations applies to the record by the LGM, but we agree that between then and ~71 ka it is much less clear what was going on.

To clarify, our argument is that there is a correlation between wet periods and human occupation before 71 ka, and a lack of correlation by ~20 ka in LGM. Humans were able to occupy diverse conditions (wet or dry) in the southern Kalahari by the LGM. Our results also suggest that the capacity to occupy this region during dry periods may predate the LGM, pending further investigations that improve as the archaeological and palaeoenvironmental records become more robust.

In summary: this is a useful paper, but I suggest the authors change their interpretation somewhat.

Thanks for the helpful feedback. We have clarified our interpretation in line with the comments made here. 

Reviewer #2

The study of von der Maden et al. concerns the study of fossil tufa deposits form the arid region of southern Kalahari in unraveling their connection with local archeological human origins and the impact for the palaeoenvironmental reconstructions.

I have appreciated the study, the obtained ages and the resulting data concerning implications with human settling

Many studies emphasize the close relation.

Thank you!

My only remarks are about:

Tufa morphological components (line 162): there are several morphological classifications about tufa depositional system. As done for the fabric classification (Manzo et al.), I suggest indicating the reference used for the morphological classification (definitions are available from Pentecost and Viles, 1994, Pedley 2009; Arenas-Abad et al. 2010; Jones and Renaut 2010, Capezzuoli et al., 2014),

As suggested by the reviewer, we have acknowledged the references drawn on for the tufa morphological classifications (line 221). We have also included additional statements in this section to clarify and credit sources for the terminology used to describe our observations, e.g. lines 266-267; 271; 284-285. 

Comparison to global records (line 271-311): in my opinion this is paragraph is a bit ambitious, not only because the here reported bibliography is really poor (please note that there is already a review article on tufa/travertine deposition and climate…and it is not cited: Ricketts, J.W., Ma, L., Wagler, A.E. and Garcia, V.H. 2019. Global travertine deposition modulated by oscillations in climate. Journal of Quaternary Science, 34, 558–568,https://doi.org/10.1002/jqs.3144), but because the tufa deposition is really different in different setting/climatic regimes of the world. I agree with authors that the dichotomy wet/dry – interglacial/glacial view is to be considered simplistic (Line 275-276), but their consequent suggestion that tufa formation is not restricted to interglacial periods is not new!

We thank the reviewer for the reference suggestion and for raising these points. We have adjusted the wording in lines 344-348 to better capture our intention with this section, which is not to evaluate global records of tufa deposition, nor to generate a one-fits-all formula for tufa formation, but rather to assess the extent to which the timing of tufa formation at Ga-Mohana can be explained by changes in selected global climate parameters, e.g. global ice volume (using the Lisiecki-Raymo benthic stack), austral summer insolation, and sea surface temperatures in the Indian Ocean.

We have included the point that tufa formation occurs across a range of climatic settings, and that their formation is controlled by a variety of factors operating on different scales, thus not merely signalling warm and wet interglacial climate conditions (lines 344-345; 353-359; 419-423). We also adjusted the wording to acknowledge that other studies report tufa formation during both glacial and interglacial periods (lines 355; 357-359).

In addition, we have expanded our interpretations of our comparisons to the selected records of climate change by explaining what exactly these comparisons mean for tufa formation at Ga-Mohana in terms of reflecting global drivers of climate change (lines 390-398 and 406-417). We conclude with a description of the conditions the Ga-Mohana tufas represent, which goes beyond the simple deposition = wet interpretation, and instead includes an acknowledgement of the influence of other parameters (evaporation rates, temperature) involved in creating favourable conditions for tufa formation in semi-arid karst settings (lines 426-438). 

For review between climate and tufa deposition, I suggest to see Andrews, 2006; Pedley, 2009, while about the former presence of active tufas as record of important rainfall regime shifts, this has been already showed in distal glacial (South Europe, Capezzuoli et al., 2010; Alexandrowicz, 2012), in semi-arid environments (Brazil, Auler & Smart, 2001; Spain, Luzon et al., 2011) and desert settings (Namibia, Viles et al., 2007; Libya, Cremaschi et al., 2010; Ethiopia, Moeyersons et al., 2006). In contrast, the presence of tufas deposits in tropical and monsoon-dominated settings testifies to an absence of destructive large wet season floods and, consequently, for reduced periods of rainfall (Carthew et al., 2003, 2006).

Thank you for this note. We are familiar with these papers and have included them as references in our new statements on lines 344-345 and 421-424, which acknowledge the multitude of studies that document the variety of settings that tufas form in, and the associated complexity of the climatic conditions they represent. 

In the correct hypothesis to study the local weather conditions (rainfall and insolation) as constraining for the local tufa deposition, I suggest to also consider and compare with the Egyptian tufa described in Kele et al. 2021 (https://doi.org/10.1144/jgs2020-147) and relative discussion implementing insolation, humidity, radiometric dating and isotope (also showing that the record of climate in Egypt’s tufa is inconsistent with a simple model of palaeoclimate for this region).

Thank you for this useful paper recommendation. We have included a paragraph (lines 382-390) to compare our observation of a lack of correlation between tufa at Ga-Mohana Hill and summer insolation to the conclusion in Kele et al 2021, which is that tufa formation in southern Egypt does not correlate to insolation either, and that the mechanisms driving rainfall in the region (and subsequent tufa formation) are complex, and cannot be attributed to a single forcing factor. 

Very minor trifles are indicated in the attached pdf (copied below):

Line 102: ‘pomeroy’ misspelled

Corrected.

Line 181: please check if the S1 Fig. citation is correct

We double-checked the caption and rearranged the order of the samples listed, otherwise it is correct.

Line 277: were. The association with interglaciation is now old!!

Corrected, we adjusted the wording to acknowledge that tufa forms during both glacial and interglacial periods (lines 353-354). 

Line 280-281: Please pay attention to the global. Noteworthy the "simple product of changing global climates" is already claimed

We have addressed this by adjusting our wording to acknowledge that our study echoes similar conclusions to those in previous studies (lines 358-358).

Line388-389: This seems an old concept.

Agree, wording adjusted. It now reads: “In accordance with other recent studies, our results challenge global generalisations of past climate change and highlight the necessity for regionally specific models.” (lines 550-552).

Line 419: Mistake or real reference??

We made a mistake with the formatting, but it’s a real reference. It has been corrected.

Hope it helps

Enrico Capezzuoli

Journal Requirements:

2. In your Methods section, please provide additional information regarding the permits you obtained to collect samples for the present study. Please ensure you have included the full name of the authority that approved the field site access and, if no permits were required, a brief statement explaining why.

Done – a statement was added to the methods section (lines 159-162).

3. Please include a complete copy of PLOS’ questionnaire on inclusivity in global research in your revised manuscript. Our policy for research in this area aims to improve transparency in the reporting of research performed outside of researchers’ own country or community. The policy applies to researchers who have travelled to a different country to conduct research, research with Indigenous populations or their lands, and research on cultural artefacts. We note that some of the authors of your manuscript are from outside of South Africa. The questionnaire can also be requested at the journal’s discretion for any other submissions, even if these conditions are not met. Please find more information on the policy and a link to download a blank copy of the questionnaire here: https://journals.plos.org/plosone/s/best-practices-in-research-reporting. Please upload a completed version of your questionnaire as Supporting Information when you resubmit your manuscript.

Done - the Inclusivity Questionnaire is included as a Supplementary Checklist and referenced in the manuscript (lines 162-164 and 977). 

4. We note that Figure 1 in your submission contain copyrighted images. All PLOS content is published under the Creative Commons Attribution License (CC BY 4.0), which means that the manuscript, images, and Supporting Information files will be freely available online, and any third party is permitted to access, download, copy, distribute, and use these materials in any way, even commercially, with proper attribution. For more information, see our copyright guidelines: http://journals.plos.org/plosone/s/licenses-and-copyright.

a) You may seek permission from the original copyright holder of Figure 1 to publish the content specifically under the CC BY 4.0 license. 

We produced the figure in ArcGIS and it contains no copyrighted images. We have added text to the Figure 1 caption to acknowledge the data sources used (lines 107-110) and updated the reference list to include these.

Done.

---

## [Decision Letter · Decision Letter 1]

6 Jun 2022

Tufas indicate prolonged periods of water availability linked to human occupation in the southern Kalahari

PONE-D-22-02819R1

Dear Dr. von der Meden,

We’re pleased to inform you that your manuscript has been judged scientifically suitable for publication and will be formally accepted for publication once it meets all outstanding technical requirements.

Kind regards,

Andrea Zerboni, Ph.D.

Academic Editor

PLOS ONE

Additional Editor Comments (optional):

We appreciate your efforts n revising your paper and we are happy to accept for publication. One of the reviewer only suggest a very recent paper that you may consider at this stage:

Kaboth-Bahr et al (2021) on the point that climate variability does in Africa does not simplistically follow a glacial/interglacial logic: https://www.pnas.org/doi/pdf/10.1073/pnas.2018277118

Reviewers' comments:

Reviewer's Responses to Questions

**Comments to the Author**

1. If the authors have adequately addressed your comments raised in a previous round of review and you feel that this manuscript is now acceptable for publication, you may indicate that here to bypass the “Comments to the Author” section, enter your conflict of interest statement in the “Confidential to Editor” section, and submit your "Accept" recommendation.

Reviewer #1: All comments have been addressed

2. Is the manuscript technically sound, and do the data support the conclusions?

Reviewer #1: Yes

3. Has the statistical analysis been performed appropriately and rigorously? 

Reviewer #1: Yes

4. Have the authors made all data underlying the findings in their manuscript fully available?

Reviewer #1: Yes

5. Is the manuscript presented in an intelligible fashion and written in standard English?

Reviewer #1: Yes

6. Review Comments to the Author

Reviewer #1: I think the authors have done a good job at revising their paper. I have no further comments and look forward to seeing this published.

The authors may wish to consider adding a reference to the recent paper by Kaboth-Bahr et al (2021) on the point that climate variability does in Africa does not simplistically follow a glacial/interglacial logic:

https://www.pnas.org/doi/pdf/10.1073/pnas.2018277118

7. PLOS authors have the option to publish the peer review history of their article (what does this mean?). If published, this will include your full peer review and any attached files.

Reviewer #1: No

---

## [Editor Report · Acceptance letter]

27 Jun 2022

PONE-D-22-02819R1 

Tufas indicate prolonged periods of water availability linked to human occupation in the southern Kalahari 

Dear Dr. von der Meden:

I'm pleased to inform you that your manuscript has been deemed suitable for publication in PLOS ONE. Congratulations! Your manuscript is now with our production department. 

Kind regards, 

on behalf of

Prof. Andrea Zerboni 

Academic Editor

PLOS ONE